# Algorithm of Femoropopliteal Endovascular Treatment

**DOI:** 10.3390/medicina58091293

**Published:** 2022-09-16

**Authors:** Maxime Dubosq, Maxime Raux, Bahaa Nasr, Yann Gouëffic

**Affiliations:** 1Department of Vascular and Endovascular Surgery, Institut Cœur-Poumon, 59000 Lille, France; 2Department of Vascular and Endovascular Surgery, Groupe hospitalier Paris St Joseph, 75014 Paris, France; 3Department of Vascular and Endovascular Surgery, Brest University Hospital, 29200 Brest, France

**Keywords:** peripheral arterial disease, endovascular procedure, femoropopliteal axis, bare metal stent, drug-eluting stent, drug-coated balloon

## Abstract

*Background and Objectives*: Indications for the endovascular treatment of femoropopliteal lesions have steadily increased over the past decade. Accordingly, the number of devices has also increased, but the choice of the best endovascular treatment remains to be defined. Many devices are now available for physicians. However, in order to obtain a high success rate, it is necessary to respect an algorithm whose choice of device is only one step in the treatment. *Materials and Methods*: The first step is, therefore, to define the approach according to the lesion to be treated. Anterograde approaches (femoral, radial, or humeral) are distinguished from retrograde approaches depending on the patient’s anatomy and surgical history. Secondarily, the lesion will be crossed intraluminally or subintimally using a catheter or an angioplasty balloon. The third step corresponds to the preparation of the artery, which is essential before the implantation of the device. It has a crucial role in reducing the rate of restenosis. Several tools are available and are chosen according to the lesion requiring treatment (stenosis, occlusion). Among them, we find the angioplasty balloon, the atherectomy probes, or intravascular lithotripsy. Finally, the last step corresponds to the choice of the device to be implanted. This is also based on the nature of the lesion, which is considered short, up to 15 cm and complex beyond that. The choice of device will be between bare stents, covered stents, drug-coated balloons, and drug-eluting stents. Currently, drug-eluting stents appear to be the treatment of choice for short lesions, and active devices seem to be the preferred treatment for more complex lesions, although there is a lack of data. *Results*: In case of failure to cross the lesion, the retrograde approach is a safe and effective alternative. Balloon angioplasty currently remains the reference method for the preparation of the artery, the aim of which is to ensure the intraoperative technical success of the treatment (residual stenosis < 30%), to limit the risk of dissection and, finally, to limit the occurrence of restenosis. Concerning the treatment, the drug-eluting devices seem to present the best results, whether for simple or complex lesions. *Conclusions*: Endovascular treatment for femoropopliteal lesions needs to be considered upstream of the intervention in order to anticipate the treatment and the choice of devices for each stage.

## 1. Introduction

In recent decades, endovascular treatment has become the first-line treatment for atheromatous lesions of the femoropopliteal (FP) segment [1]. Already in 2009, authors observed a decline in the number of revascularizations in open surgery in favor of an increase in endovascular procedures [2]. In 2019, nearly 50,000 patients in France benefited from FP revascularization by the endovascular procedure. Over the past twenty years, indications for endovascular treatment compared to surgical treatment have evolved. With respect to the TASC-II classification from 2000, endovascular treatment was the preferred approach for femoropopliteal stenosis ≤ 5 cm or occlusions ≤ 3 cm. A few years later, the TASC-II guidelines published in 2007 recommended endovascular repair for lesions of ≤15 cm [3]. The latest recommendations published in 2017 by the European Society for Vascular and Endovascular Surgery recommended endovascular treatment for femoropopliteal lesions up to 25 cm in length [1]. In parallel, technical advances have led to the widespread applicability of endovascular repair to improve outcomes and treat more challenging femoropopliteal lesions. Most of these technical advances, such as retrograde puncture, atherectomy, chronic total occlusion (CTO) guidewires, or drug-eluting therapies, are currently part of our armentorium. However, for a long-time, the algorithm for the femoropopliteal segment was just reduced to the choice of an implantable device. However, many steps are crucial to obtain a high technical procedure success and to improve outcomes.

Herein, we propose an algorithm for the endovascular treatment of atheromatous femoropopliteal lesions. This algorithm is based on four stages that include approach, crossing, preparation, and the treatment of the lesion (Figure 1).

## 2. Define the Approach

The choice of approach depends on the location of the lesions requiring treatment. Anterograde ultrasound-guided puncture of the common femoral artery (CFA) can be performed in cases of femoropopliteal lesions sparing the proximal third of the FP segment and in the absence of associated iliac or CFA ipsilateral lesions. The main advantages of this puncture are the direct and rapid approach to the lesion, better pushability, and the use of shorter catheters and guidewires. Its main disadvantages are the direct irradiation of the hands located under the X-ray beam at the time of catheterization of the superficial femoral artery (SFA) and the risk of retroperitoneal hematoma after compression [4,5]. A variant of the antegrade CFA puncture is the direct superficial femoral artery puncture which appears to be safe [6]. The contralateral retrograde puncture requires mastering the crossover technique and using long catheters (>120 cm) (Figure 2). Crossover should be avoided in the event of significant aorto–iliac lesions, severe iliac tortuosity, calcified iliac arteries, and the presence of bypass or aorto-bi-iliac endograft.

More rarely, other approaches are described for the treatment of femoropopliteal lesions. The *Precise Retrograde Supera Stenting of the Ostium* (PRESTO) technique consists of performing a retrograde puncture of P1 or the distal third of the superficial femoral artery and treating the lesions using this approach [7]. The authors recommend performing the procedure without a sheath. Some authors have also reported the use of upper limbs approaches (radial or brachial) [8,9]. These studies report the feasibility of these approaches but observe significant local and neurological complication rates. For others, open brachial access could be considered a safe and secure alternative approach for patients when femoral artery access is unavailable [10]. Finally, some authors have evaluated the leg or even pedal approach as a first-line approach to treat infra-inguinal lesions [11,12].

In the majority of procedures, a 6 Fr sheath is used to perform the procedure. Some physicians prefer 4–5 Fr sheaths in order to limit complications at the puncture site, and others prefer 7 Fr sheaths in order to use atherectomy systems. In the case of a contralateral puncture, a long (45 cm) and braided sheath should be preferred. Once the common femoral is in place, arteriography of the lower limb must be performed to identify the arterial tree from the puncture point to the foot. This examination can be performed by staged injections of contrast product or by carrying out a bolus of contrast product synchronized with a translational movement of the table (bolus chase). Thanks to arteriography, the operator can visualize all the arterial lesions of the lower limb and compare them with the results of the preoperative morphological examination.

## 3. Crossing the Lesion

In case of stenosis, a navigation wire can be enough to cross the lesion. A support catheter must be used in order to undo a loop of the wire which could cause an unwanted dissection. Platforms 0.035 or 0.018 are currently most often used. The support catheter can be a dedicated catheter or a regular balloon angioplasty catheter. The angioplasty balloon catheter has the advantage of being able to begin the preparation of lesions while reducing resistance to catheter progression when lesions are extensive. In the event of CTO lesions, a crossing strategy must be considered. Currently, there is no evidence in favor of the intraluminal or subintimal route in terms of mid-term results for bare metal stent or drug-eluting devices [13,14,15,16]. The subintimal navigation is performed by forming a loop with a hydrophilic wire and with catheter support. (Figure 3) The loop must be undone at the level of the reentry zone that the physician will have determined. If reentry into the true lumen has not been possible using conventional wire and catheter techniques, an Outback^®^ (Cordis) or Goback^®^ (Upstream peripheral technology) type reentry device can be used. The subintimal route presents a contraindication to the use of most atherectomy systems. In all cases, reentry into the lumen of the target artery must be controlled. For this purpose, the support catheter is positioned a few centimeters downstream of the supposed reentry location, the guidewire is withdrawn, and a few milliliters of contrast product are injected into the lumen of the support catheter.

The estimated failure rate of crossing a femoropopliteal lesion by the anterograde route is 10–20% [17]. In the case of failure, a retrograde approach can be performed in order to increase the chances of crossing the lesion. The retrograde approach can be performed by puncturing the distal third of the superficial femoral artery, the popliteal, or the below-knee arteries (Figure 4). When the two wires are located in the same subintimal space, it is called subintimal arterial flossing with antegrade–retrograde intervention (SAFARI) [18]. When the guidewires are not in the same plane, two undersized balloons are advanced via the antegrade and retrograde pathways, placing their extremities at the same level without overlapping the two balloons. Both balloons are inflated for a few seconds to break up the plaque or dissection plane and allow the two planes to connect. A technique of retrograde puncture of the pedal artery without the use of an introducer has also been described for the treatment of popliteal lesions [19]. In a meta-analysis, *Giannopoulos* et al. report a technical success rate of retrograde puncture of 96% associated with a low rate of complications (Perforation: 2.1%; flow-limiting dissection: 0.6%; distal embolization: 0.1% and hematoma at retrograde puncture: 1.3%) [17]. At the end of the procedure, the hemostasis of the retrograde approach could be achieved by manual compression or balloon angioplasty.

## 4. Arterial Lesion Preparation

The preparation of the lesions has recently become an essential part of the treatment algorithm for the femoropopliteal segment. There are multiple objectives for vessel preparation. First, the vessel preparation should ensure the intraoperative technical success of the treatment (residual stenosis < 30%) to limit the risk of dissection and, finally, to limit the occurrence of restenosis [20]. However, the evidence for the effectiveness of arterial preparation devices is weak, regardless of the type of arterial preparation device.

### 4.1. Balloon Angioplasty 

The interest in arterial preparation by angioplasty balloon has been shown by several studies (Figure 5). The randomized Zilver PTX trial showed a trend towards better patency at 5 years in the drug-eluting-stent (DES) and preparation group versus the DES without preparation group (72.4 vs. 64.9%) [21]. On the other hand, *Zorger* et al. studied the influence of the duration of the preparation of the artery by balloon [22]. In this study, the prolonged inflation of 180 s versus 30 s makes it possible to significantly reduce the rate of major dissection. However, it has been shown that dissections are negatively correlated with the rate of patency and the absence of reintervention at the level of the target lesion [23]. The choice of the diameter of the angioplasty balloon depends on the treatment. For example, for certain low chronic outward force bare metal stents (BMS), it is recommended to choose a balloon that is slightly oversized compared to the nominal arterial diameter. Conversely, the diameter of the balloon which prepares the artery before drug-coated-balloon (DCB), BMS or DES must be undersized by 1 mm compared to the nominal diameter of the target artery [24,25,26,27,28]. In addition, the technical characteristics of the predilation, such as the duration, the inflation pressure, or the diameter of the balloon, are rarely specified in the studies [25].

### 4.2. Dedicated Arterial Preparation Balloons: Scoring Balloon, Cutting Balloon, Chocolate Balloon

Cutting balloons are mainly indicated in cases of fibrous plaque resistant to angioplasty, and their use is mainly observed in the treatment of hemodialysis access. The scoring balloon consists of microblades located on the surface of a semi-compliant balloon. There are different designs with helicoidal (AngiosculptTM^®^; Philips, San Diego, CA, USA) or longitudinal (UltrascoreTM^®^; BD Interventional, Tempe, AZ, USA) nitinol wires. To date, there is no proof of the superiority of the scoring balloon vs a standard balloon. The chocolate balloon technology allows the maintenance of a constant diameter thanks to a nitinol mesh body over the entire length of the balloon, thus avoiding the “dog bone” effect and, therefore, the risk of dissection at the proximal or distal ends. Similarly, there is no evidence of the superiority of this technology compared to a standard balloon.

### 4.3. Atherectomy Catheters

As in open surgery, atherectomy aims to destroy and/or remove atheromatous plaque. Atherectomy can be directional (SilverHawk^®^, TurboHawk^®^, Hawkone^®^, Pantheris^®^), orbital (Diamondback360 R^®^ Peripheral Orbital Atherectomy), rotative (Jetstream^®^, Phoenix^®^), or laser (Turbo-Elite^®^ laser atherectomy catheter). Two randomized trials evaluating atherectomy have been published. A first study compared the treatment of femoropopliteal lesions that ranged from 7 to 15 cm in length by DCB with or without directional atherectomy (SilverHawk^®^, TurboHawk^®^) [29]. The authors found no difference in terms of stenosis and reintervention at the level of the target lesion at one year. In a second study comparing simple balloon preparation versus orbital atherectomy (Diamondback360 R^®^ Peripheral Orbital Atherectomy), the authors found no difference in terms of restenosis and target lesion reoperation at 1 year [30].

### 4.4. Intravascular Lithotripsy

The Intravascular Lithotripsy Catheter (IVL^®^; Shockwave Medical, Santa Clara, CA, USA) includes a dedicated balloon catheter combined with miniaturized, arrayed lithotripsy emitters to create a localized field effect that passes through the vessel wall. The waves selectively crack intimal and medial calcium. After a few cycles, the balloon can prepare the lesion with low pressure. Recently the first results of a randomized study have been published [31]. This study compares an arterial preparation of moderately or severely calcified femoropopliteal lesions by IVL versus balloon angioplasty. The primary endpoint of this study is a technical success, defined as ≤30% inferior residual stenosis without flow-limiting dissection. This study found greater technical success in the intravascular lithotripsy group (65.8% vs. 50.4%; *p* = 0.01) with no difference at one month in terms of reoperation at the target lesion between the two groups (0.7% vs. 0.7%; *p* = 1.0). Additionally, secondary endpoints were published. In this release, the primary patency at 1 year was superior in the IVL group compared to the PTA group (80.5% vs. 68.0%, *p =* 0.017), but the difference in primary patency was driven by the freedom from provisional stent placement rate [32]. Moreover, freedom from the individual endpoints of clinically driven target lesion revascularization and restenosis at 1 year were similar between the two groups.

## 5. Treatment

The treatment of femoropopliteal lesions can be divided into two groups depending on the length of the lesion: short lesions < 15 cm and long lesions ≥ 15 cm. Indeed, most randomized trials comparing endovascular treatment to open surgery for long femoropopliteal lesions have defined these lesions as being greater than ≥15 cm [33,34,35]. It is important to ensure that a prepared area is completely treated in order to avoid geographic miss. For this, the length of the device must be greater than the prepared area (0.5 to 1 cm at each end). Regarding the diameter, the ratio between the treatment device and the nominal diameter must be 1:1.

### 5.1. Short Lesions

Currently, endovascular treatment is the first line of treatment for short femoropopliteal lesions [1]. Historically the treatment of femoropopliteal lesions was balloon angioplasty, and stenting was indicated in case of the failure of angioplasty [36]. During the 2000s, various studies showed the superiority of bare metal stents over balloon angioplasty in terms of patency and clinical improvement [37,38,39]. In the absence of evidence, a class effect is considered to exist for all bare metal stents (including those considered to have low chronic outward force). More recently, randomized controlled trials have shown the advantage of paclitaxel-eluting devices over angioplasty in terms of patency [21,25]. Nowadays, angioplasty is no longer considered a routine treatment for lesions of the femoropopliteal segment.

Currently, two paclitaxel-eluting active nitinol stents are available: Zilver PTX^®^ (Cook, Bloomington, IN, USA) without a controlled release of paclitaxel (concentration: 3 µg/mm^2^) and the Eluvia^®^ stent (Boston Scientific, Marlborough, MA, USA) with controlled release of paclitaxel (PVDF-HFP polymer) (concentration: 0.167 µg/mm^2^). In the randomized BATTLE trial, Zilver PTX^®^ failed to show a significant reduction in in-stent restenosis at 1 year compared to a bare nitinol stent [40]. Conversely, in the d EMINENT randomized controlled trial, comparing the efficacy of active stent Eluvia^®^ versus BMS commonly used on the European market, the primary patency at 1 year was significantly higher in the Eluvia^®^ group compared to BMS (83.2% vs. 74.3%; *p* = 0.0077) (Gouëffic, Viva, 2021). Finally, the IMPERIAL randomized study, comparing Eluvia active stent and Zilver PTX^®^ stent, showed better primary patency at 1 year in favor of Eluvia^®^ (86.8 vs. 77.5% *p* < 0.0144) [41].

Different drug-coated balloons are currently available such as In.Pact Admiral^®^ (Medtronic, Dublin, Ireland), Sequent Please OTW^®^ (B Braun Medical), Ranger^®^ (Boston Scientific, Malborough, MA, USA), Luminor^®^ (Ivascular, Vascular SLU, Barcelona, Spain), Lutonix^®^ (Bard-BD, Tempe, Arizona), and Stellarex^®^ (Philips). To date, the best formulation remains unknown because no direct comparison between different drug-coated balloons has been published. A dose–response relationship was also observed in a meta-analysis [42]. However, due to confounding factors, particularly the complexity and preparation of the lesions, no causal link could be established.

There is no direct comparison between drug-coated balloons and bare metal stents. A network meta-analysis compared these two devices, and it appears that BMS is a satisfactory substitute for the drug-coated balloons with the absence of a significant difference at 2 years in terms of restenosis and reintervention [43]. Finally, two randomized studies have compared drug-coated balloons and drug-coated stents [44,45]. REAL PTX compared In.Pact Admiral^®^, In.Pact Pacific^®^, Lutonix^®^) with Zilver PTX^®^. This study was underpowered, and no difference was noted between both devices in terms of patency for short or medium-length lesions [44]. Drastico also compared drug-coated balloons versus Zilver PTX^®^ [45]. Drug-coated balloons were not superior to Zilver PTX^®^ in the treatment of complex FP lesions in a high-risk population. However, both studies assessed a polymer-free drug-eluting stent, and comparisons with polymer paclitaxel-eluting stents are still expected.

Finally, more and more studies are assessing the safety and efficacy of limus-eluting devices. Head-to-head comparisons of limus-eluting devices with paclitaxel-eluting devices are warranted [46,47].

### 5.2. Complex Lesions

The latest ESVS guidelines propose a first-line endovascular approach for femoropopliteal lesions up to 25 cm long [1]. Beyond that, open surgery remains recommended if the patient is compatible with this procedure and if the saphenous vein is available. Recently, three randomized clinical trials have compared endovascular treatment with open surgery for the treatment of long femoropopliteal lesions [33,34,35]. The results of these studies suggest that a primary endovascular strategy using DES, BMS, or covered stent is a reasonable alternative to surgery. However, to date, the best endovascular treatment for long femoropopliteal lesions remains to be defined. In a recent meta-analysis (Dubosq, Société Française de Chiriurgie Vasculaire et Endovasculaire, 2022), analyzing 44 studies and 4847 patients treated for femoropopliteal lesions longer than 150 mm, we studied the patency rates of the different devices. The average lengths of the lesions varied between 150.5 mm and 330 mm. The primary patency and reintervention rate of the target lesion at 1 year, all devices combined, were 71% and 21%. The patients treated with drug-coated balloons and drug-eluting stents had the best patency rates at 1 year: 0.74 [95% CI: 0.64–0.84] and 0.83 [95% CI: 0.78–0.88], respectively. All of these data could enlarge the indication of endovascular treatment for all femoropopliteal lesions, whatever the length.

### 5.3. Paclitaxel Safety

In 2018, *Katsanos* et al. published a meta-analysis that raised doubts about the safety of using eluting devices at the femoropopliteal level [48]. Since 2018, numerous publications have studied this risk and did not find any risk of excess mortality described. In 2019, a two-year update of the meta-analysis by *Katsanos* et al. relating to the specific analysis of drug-eluting stents did not find a mortality signal demonstrating the fragility of the signal [49]. Different national cohort studies conducted in the United States and in Germany did not find any excess mortality in patients treated with eluting devices [50,51,52]. The VOYAGER PAD randomized controlled trial did not find any significant difference in terms of mortality between the groups with and without an active device [53]. More recently, *Nordanstig* et al. published the results of an interim analysis of the SWEDEPAD study and found no significant difference in terms of survival between the groups with or without an eluting device for the treatment of the femoropopliteal segment [54]. Currently, the accumulated evidence diminishes any doubt about paclitaxel safety, and we have more and more data about paclitaxel efficacy when paclitaxel-eluting devices are used as the first line of treatment for femoropopliteal lesions.

## 6. Conclusions

The endovascular management of femoropopliteal lesions requires a precise morphological assessment in order to plan the different methods of approach, crossing, preparation, and treatment of the lesions. The choice of treatment for the target lesion is oriented towards drug-eluting devices for the shortest lesions (<15 cm), while for the longest lesions (≥15 cm), the type of treatment remains to be defined.

## Figures and Tables

**Figure 1 medicina-58-01293-f001:**
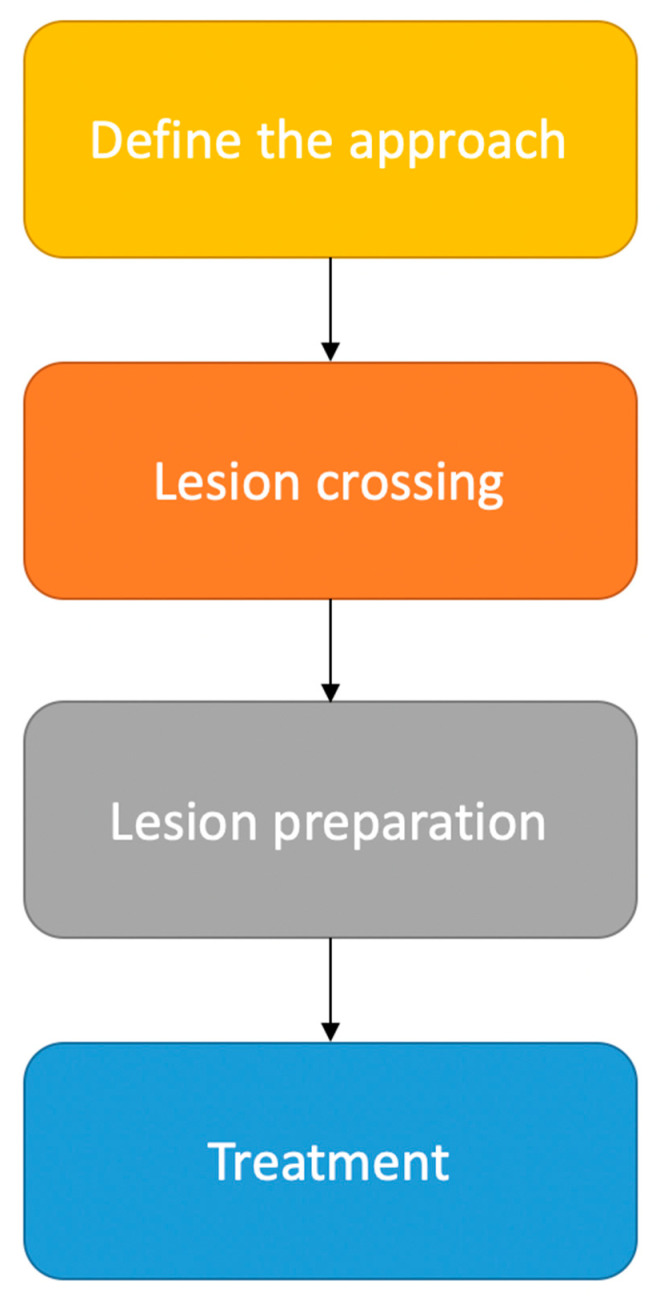
Strategy for endovascular treatment of femoropopliteal lesions.

**Figure 2 medicina-58-01293-f002:**
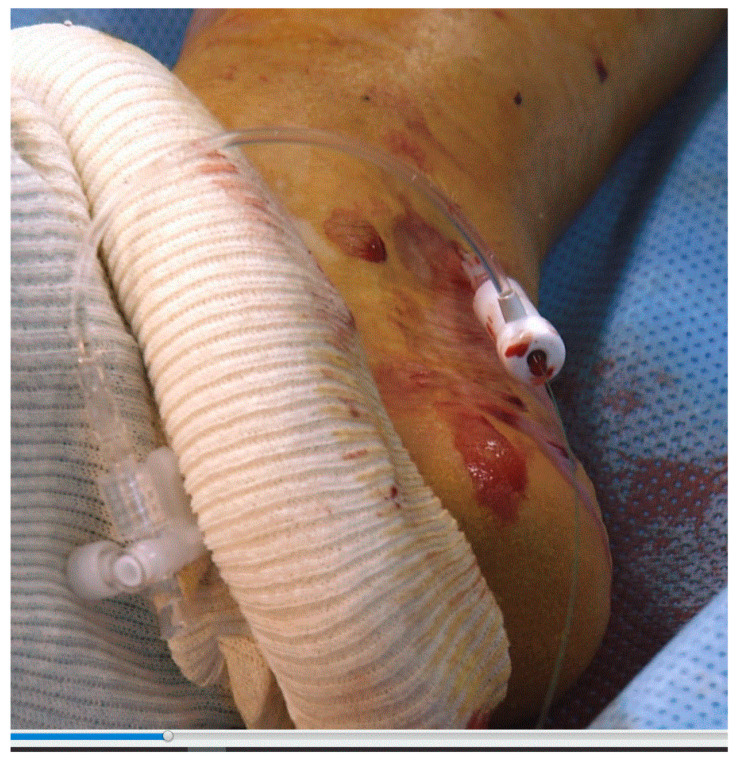
Contralateral posterior tibial artery puncture.

**Figure 3 medicina-58-01293-f003:**
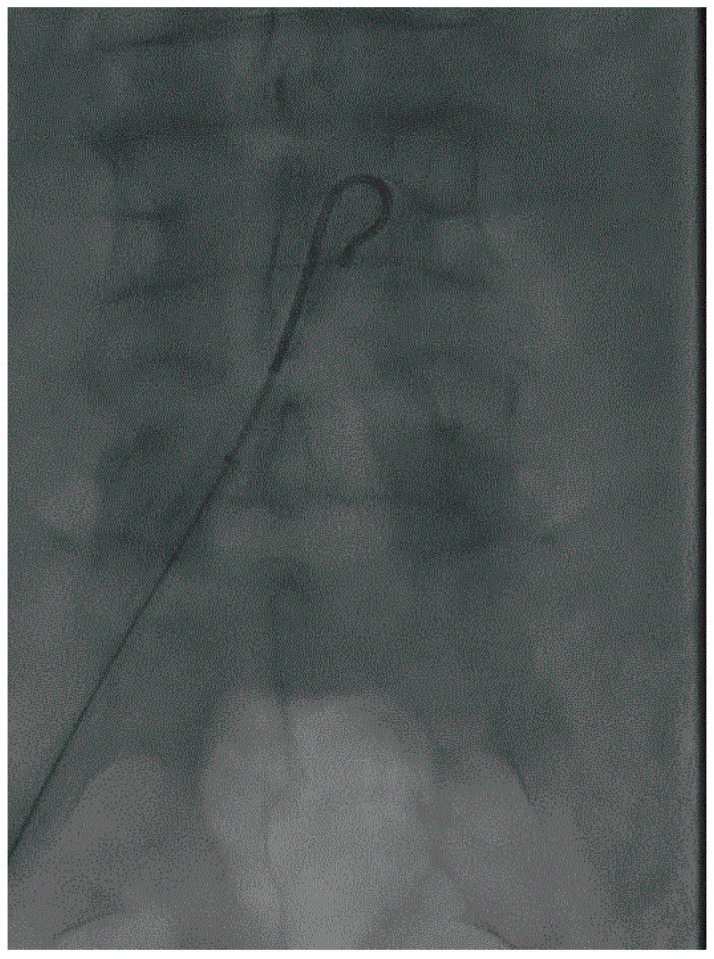
Crossover technique. 6F-45 cm sheath, UF catheter and hydrophilic guidewire are associated to realize the crossover.

**Figure 4 medicina-58-01293-f004:**
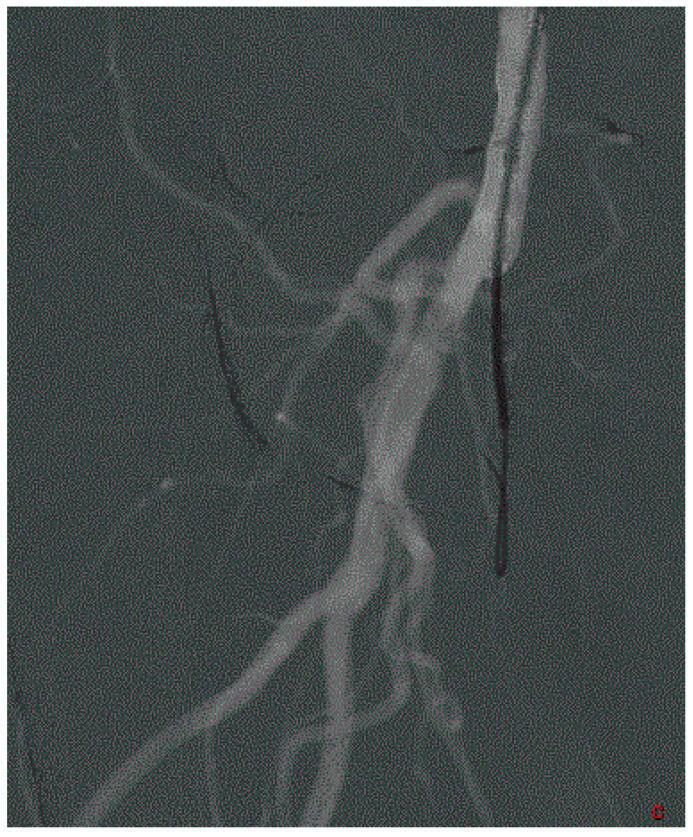
Subintimal recanalization of a long femoropopliteal occlusion with the loop technique.

**Figure 5 medicina-58-01293-f005:**
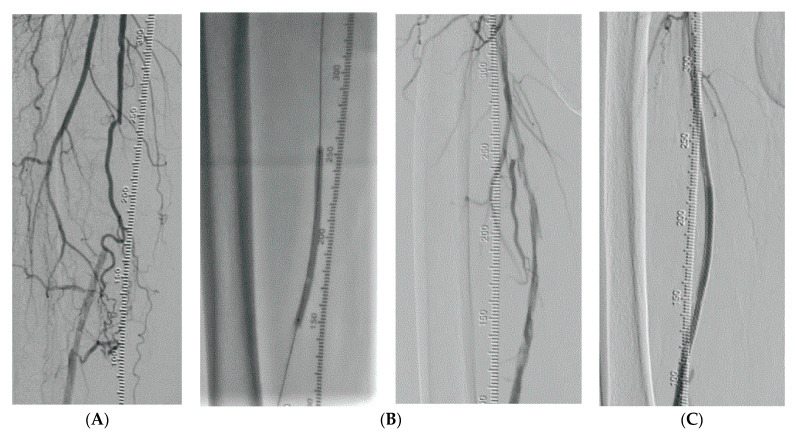
Vessel preparation and treatment. (**A**) Femoropopliteal occlusion. (**B**) Vessel preparation and control. (**C**) Final result with stenting.

## Data Availability

Not applicable.

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
