# Peer review of "Algorithm of Femoropopliteal Endovascular Treatment"

_medicina, 2022, doi:10.3390/medicina58091293_

Round 1
Reviewer 1 Report
medicina-1856654
The authors propose an algorithm for the endovascular treatment of femoropopliteal lesions. This review is well written overall, despite only a few shortcomings. It gives a profound and clear overview of the state-of-the-art treatment options.
Major
1. Figure1 (page 3): The amount of information in this figure is little. I would suggest to either reduce the size of the figure or to add information in terms of possible devices.
2. (page 4, line 89-90): The significant local complications refer to a percutaneous approach, this should be mentioned as well as the possibility of a surgical transbrachial cut-down, which is much safer.
3. (page 4, line 114): The references 10 and 11 refer to stenting only, what about PTA/DEB? Can you please add the information or at least comment?
4. There appears to be a problem with the arrangement of the citation from 12 onwards. Can you please correct this?
5. (page 5, line 138): At the end of the procedure, how to deal with the retrograde puncture site? Can you please add this information to the paragraph?
6. (page 7, line 254): Where is the algorithm? POBA, DEB, BMS, DES? Can you please add this information tot he paragraph. What about Sirolimus-coated balloon, can you please comment on that?
7. (page 8, line 287): The paragraph lacks a conclusion.
Minor
1. In general: Please be consistent in terms of the description of lesion length. 5 cm or 5cm (written together or not).
2. Introduction (page 2, line 47-49): Please rephrase the sentence, e.g. “With respect to the TASC-II classification from 2007 endovascular treatment was …”
3. (page 2, line 51): change “lesion femoropopliteal” to “femoropopliteal lesion”
4. (page 2, line 55-56): Change to “However, during a long-time the algorithm for the femoropopliteal segment was just reduced to the choice of an implantable device”.
5. (page 4, line 77-78): Please provide citations for “increased radiation for hand” and “risk of retroperitoneal hematoma”.
6. (page 4, line 112): Please delete “complete and chronic”, this is redundant.
7. (page 4, line 116-117): “…failure into the true lumen, …”
8. (page 5, line 144): “residual stenosis < 30%”. Can you please add a citation?
9. (page 6, line 174: “… compared to a standard balloon.”
10. (page 6, line 202): CD-TLR, was it defined before?
11. (page 7, line 223): “… bare metal stents…”
Author Response
Reviewer 1
medicina-1856654
The authors propose an algorithm for the endovascular treatment of femoropopliteal lesions. This review is well written overall, despite only a few shortcomings. It gives a profound and clear overview of the state-of-the-art treatment options.
Major
- Figure1 (page 3): The amount of information in this figure is little. I would suggest to either reduce the size of the figure or to add information in terms of possible devices.
I would suggest to reduce the size and not to add info about devices, just to keep attention on the main steps.
- (page 4, line 89-90): The significant local complications refer to a percutaneous approach, this should be mentioned as well as the possibility of a surgical transbrachial cut-down, which is much safer.
We thank the reviewer for this question. The text was altered to:
“Some authors have also reported the use of upper limbs percutaneous approaches (radial or brachial)6,7. These studies report the feasibility of these approaches but observe significant local and neurological complication rates. For others, open brachial access could be considered as a safe and secure alternative approach for patients when femoral artery access is unavailable.(Nasr, Ann Vasc Surg, 2016)
- (page 4, line 114): The references 10 and 11 refer to stenting only, what about PTA/DEB? Can you please add the information or at least comment?
We thank the reviewer for this comment. For DCB, patency rates seems also similar after intraluminal versus subintimal recanalization. (Schmidt, Jacc Intv 2016, Micari, Jacc Intv, 2013). References were added and the text was modified to : « Currently there is no evidence in favor of the intraluminal or subintimal route in terms of medium-term results for bare metal stent or drug eluting devices.(Ishihara ; J Endovasc Ther 2016 ; Soga Y, J Vasc Surg 2013 ; Schmidt, Jacc Intv 2016, Micari, Jacc Intv, 2013)
- There appears to be a problem with the arrangement of the citation from 12 onwards. Can you please correct this?
Accordingly to this comment, all the references have been revised.
- (page 5, line 138): At the end of the procedure, how to deal with the retrograde puncture site? Can you please add this information to the paragraph?
We thank the reviewer for this question. At the end of the procedure, the hemostasis of the retrograde approach could be achieved by manual compression or balloon angioplasty. The text was modified accordingly.
- (page 7, line 254): Where is the algorithm? POBA, DEB, BMS, DES? Can you please add this information tot he paragraph. What about Sirolimus-coated balloon, can you please comment on that?
Herein, we did not want to define any algorythm for the treatment. Indeed, level one evidences are still missing to establish this kind of algorytm. For example head to head comparisons between BMS and DCB does not exist. Moreover, no head to head comparison exist betweem polymer paclitaxel eluting stent and DCB.
The text was altered accordingly.
- (page 8, line 287): The paragraph lacks a conclusion.
Accordingly, we have added a conclusion.
All these data could enlarge the indication of endovascular treatment to all femoropopliteal lesions, whatever the length.
Minor
- In general: Please be consistent in terms of the description of lesion length. 5 cm or 5cm (written together or not).
Revised
- Introduction (page 2, line 47-49): Please rephrase the sentence, e.g. “With respect to the TASC-II classification from 2007 endovascular treatment was …”
Thank you for this comment. The text was altered to : «With respect to the TASC-II classification from 2000, endovascular treatment was the prefered approach for femoropopliteal stenosis ≤ 5cm or occlusions ≤ 3cm. Few years later, the TASC-II guidelines published in 2007 recommanded endovascular repair for esions ≤15cm. »
- (page 2, line 51): change “lesion femoropopliteal” to “femoropopliteal lesion”
Done.
- (page 2, line 55-56): Change to “However, during a long-time the algorithm for the femoropopliteal segment was just reduced to the choice of an implantable device”.
Done
- (page 4, line 77-78): Please provide citations for “increased radiation for hand” and “risk of retroperitoneal hematoma”.
For « increased radiation for hand », we have choosen : C. Nice, CardioVascular and Interventional Radiology 2003.
For « risk of retroperitoneal hematoma”, we have choosen : Kentaro Fukuda Plos One 2021 (https://doi.org/10.1371/journal.pone.0248416)
- (page 4, line 112): Please delete “complete and chronic”, his is redundant.
Done
- (page 4, line 116-117): “…failure into the true lumen, …”
The sentence has been altered to : « If reentry into the true lumen has not been possible using conventional wire and catheter techniques, an Outback® (Cordis) or Goback® (Upstream peripheral technology) type reentry device can be used.
- (page 5, line 144): “residual stenosis < 30%”. Can you please add a citation?
A reference was added to the text : Diehm Eur J Vasc Endovasc Surg 2008
- (page 6, line 174: “… compared to a standard balloon.”
Done
- (page 6, line 202): CD-TLR, was it defined before?
CD-TLR was spelled in the text as clinically driven target lesion revascularization
- (page 7, line 223): “… bare metal stents…”
Done

Reviewer 2 Report
Major Comments:
1. A table and additional figure or model would add to the impact of the work.
2. Additional discussion about how this model would improve upon current practices already in place would add to the manuscript. As presented, the manuscript appears to be more of a review rather than a research article.
3. Are there examples of successful implementation of this model in the clinical practice?
Author Response
Reviewer 2
Major Comments:
- A table and additional figure or model would add to the impact of the work.
According your comment, 4 figures were added to the manuscript :
Figure 2 : Crossover technique
Figure 3: Subintimal recanalization with the loop technique
Figure 4 : Below the knee retrograde appraoch for SAFARI revascularization
Figure 5 Vessel preparation and treatment
- Additional discussion about how this model would improve upon current practices already in place would add to the manuscript. As presented, the manuscript appears to be more of a review rather than a research article.
In fact, you are right. This paper is more a review than a research article or a model. Indeed, according the current practice of vascular interventionalists, we have described 4 steps of the treatment of the femoropopliteal treatment.
- Are there examples of successful implementation of this model in the clinical practice
See above.

Round 2
Reviewer 2 Report
Thank you for your edits. I have no additional comments